# Seismic Response of Acceleration-Sensitive Non-Structural Components in Buildings

Carmine Lima  and Enzo Martinelli *

Department of Civil Engineering, University of Salerno, 84084 Fisciano (SA), Italy; clima@unisa.it
* Correspondence: e.martinelli@unisa.it; Tel.: +39-089-964098

**Abstract:** This paper aims at highlighting the main mechanical parameters controlling the behavior of the so-called 'acceleration-sensitive' non-structural components (NSCs). The first reports a short review of the current state of knowledge and the critical issues dealing with the prediction of the seismic response of NSCs. Then, the paper presents the results of a numerical parametric analysis intended to capture the key features of the coupled dynamic response of a two-degree-of-freedom (2DOF) system supposed to be representative of both main structure and 'non-structural' component (NSC). The main parameters controlling the dynamic response of NSCs emerge from this study, which could pave the way towards formulating more mechanically consistent relationships for evaluating the peak accelerations induced by seismic shakings on NSCs in buildings.

**Keywords:** seismic analysis; non-structural components; nonlinear analysis; 2DOF; maximum acceleration

## 1. Introduction

Significant research efforts have been produced in recent decades with the aim to formulate sound criteria for the design of structures in seismic areas resulting in the current generation of seismic codes and guidelines [1]. Such codes provide designers with consistent performance-based approaches for designing and assessing structures against earthquake-induced actions, although more comprehensive and reliability-based approaches [2] could be implemented in future evolutions. However, a series of critical issues, not completely addressed by the current codes, emerge by analyzing damage suffered from existing structures in recent earthquake events [3]. Specifically, inaccurate predictions of the seismic response of "non-structural components" (NSCs) [4–6] have emerged, for instance, in the aftermath of the event occurred in Emilia Region, Italy [7], where several precast buildings mainly suffered damage related to inadequate design of connections between structural members and NSCs [8,9]. Therefore, predicting the seismic response of NSCs is perceived as one of the most important challenges of the seismic engineering research community [10,11].

Several definitions for the very wide class of 'objects' often referred to as NSCs are available both in the scientific literature and in recent seismic codes [2]. Generally, any 'object' that does not contribute to support both gravity and seismic actions in the model considered in structural analysis is considered a 'non-structural' or 'secondary' element. As matter of fact, masonry infill, partitions, finishing, suspended ceilings, as well as dedicated equipment needed for the specific purpose of the construction under consideration are the most common NSCs in buildings.

Moreover, recent scientific research and technical codes introduced further definitions and classifications of NSCs: a review of these definitions is available in the literature [12]. Generally, they are based on different aspects, such as the component's purpose or function, the way in which it is actually connected to the main structure and the sensitivity to particular aspects (acceleration, displacement, and so on) of the dynamic response.

Over the classification of NSCs, the main objective of various seismic codes in force in earthquake-prone countries (e.g., [1,13–16]) is to evaluate the maximum acceleration, and thus the maximum inertial force, on NSC induced by the expected seismic shaking. However, rules and relationships provided with this purpose are generally simple (not to say simplistic) and disregard fundamental parameters which could significantly affect the dynamic response of NSCs.

As matter of principle, rules and relationships currently provided involve few parameters dealing with the intensity of expected earthquakes, the elastic parameters of both main structure and NSC and the position in elevation of the NSCs as part of the main structure. A thorough discussion about the limitation of code formulations has been recently proposed [12]; specifically, it emerges that the analyzed code-provisions either disregard or not explicitly consider the nonlinear behavior of the main structure that may clearly affect the excitation of the NSCs by filtering the input seismic signal [17].

Therefore, this paper presents a wide parametric analysis based on a two-degree-of-freedom (2DOF) system used for simulating the dynamic response of a general structure with generic "acceleration-sensitive" NSCs [18]. It should be intended as a further development of previous study [12]. In fact, the latter mainly proposed a wide state-of-the-art report about the historical evolution of code provisions for "acceleration-sensitive" NSCs; then, it presented a preliminary parametric analysis demonstrating that the current EC8 [1] formula fails in predicting the maximum seismic-induced acceleration on those components, as it misses considering some relevant parameters. Therefore, the present paper is characterized by the following novel elements:

- it recalls the definition of two alternative response parameters for describing the response of "acceleration-sensitive" NSCs;
- it shows the how several mechanical properties, currently neglected or disregarded by code provisions, actually affect the average value of the maximum inertial force acting on NSCs;
- it proposes some preliminary considerations about how the record-to-record variability of the generic "acceleration-sensitive" NSCs under consideration.

Section 2 summarizes the main features of a 2DOF system intended at simulating the interaction between "acceleration-sensitive" NSCs and main structure: the model is utilized for executing a parametric analysis intended at covering a wide set of natural seismic records and structures characterized by variable values of the relevant parameters. The most relevant results of the proposed parametric analysis are summarized in Section 3: they mainly aim at revealing the relevant parameters affecting the maximum seismic-induced accelerations on NSCs. It is worth highlighting that this study does not cover the case of "deformation-sensitive" NSCs (such as the in-plane behavior of masonry infills and partition walls) [18], whose mechanical interaction with the main structure needs to be described through different parameters [19].

## 2. Parametric Investigation

The interaction affecting the dynamic response of the main structure and the NSC connected to it is investigated by defining a two-degree-of-freedom (2DOF) system, which is considered as the simplest possible representation of the dynamic problem under consideration: Figure 1 represents such a system as it is utilized in the present study.

An elastic-perfectly-plastic behavior is supposed for the main structure. It is characterized by elastic stiffness $k$, mass $m$, viscous damping $c$ and yielding force $F_y$ (Figure 1). The parameters $x$ and $\dot{x}$ denote the relative displacement and velocity of the main structure with respect to the ground, respectively. The NSC is represented by its mass $m_a$ and it is connected to the main structure by an elastic element with stiffness $k_a$. The relative displacement of the NSC with respect to the ground is denoted with $x_a$. The viscous damping coefficient $c_a$, which relates the viscous force with the relative velocity $\dot{x}_a$ of the NSC with respect to the ground, completes the description of the 2DOF system under investigation.

However, nonlinear behavior is not considered for the NSC in this study, since it is mainly devoted to evaluating the maximum forces induced on secondary components without covering aspects related to displacements.

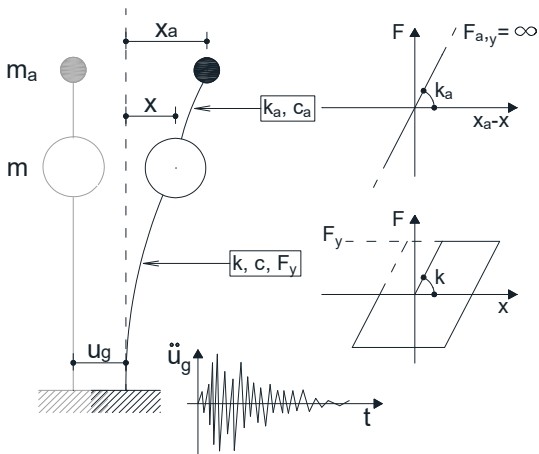

**Figure 1.** 2DOF system for nonlinear dynamic (time-history) analyses.

The system represented in Figure 1 allows to consider the coupled behavior of the main structure and NSC and can result more appropriate than systems generally adopted in similar studies which are often based on the dynamic analysis of two uncoupled single-degree-of-freedom (SDOF) systems in series [20,21]. As matter of fact, the latter systems are based on the simulation of a SDOF system representing the main structure whose response is, subsequently, considered as the ground motion for the secondary SDOF system which simulates the NSC. Such a study can result in accurate prediction if the NSC-to-structure mass ratio is quite small (i.e., $m_a/m{\to}0$) and thus the mass $m_a$ does not influence the dynamic response of the main structure. For sake of generality, the present study does not consider this approximation and a system of coupled equilibrium equations is actually solved by means of a piecewise approach based on the Newmark-beta numerical algorithm [22]. This algorithm is used in order to handle nonlinearities in the dynamic response of the system

$$\begin{cases} m\ddot{x} + c\dot{x} - c_a\left(\dot{x}_a - \dot{x}\right) - k_a(x_a - x) + F_r\left(x; k, F_y\right) = -m\ddot{u}_g \\ m_a\ddot{x}_a + c_a\left(\dot{x}_a - \dot{x}\right) + k_a(x_a - x) = -m_a\ddot{u}_g \end{cases} \tag{1}$$

In Equation (1) the reaction $F_r(x,k,F_y)$ is the unique nonlinear part which includes both the relative displacement $x$ of the main structure and its stiffness $k$ and yielding force $F_y$ (Figure 1).

A set collecting 264 natural seismic records has been employed as ground motion in the nonlinear time-history analyses of the 2DOF system described above carrying out a very wide parametric analysis. This set collects natural records considered in a significant study devoted to investigating the nonlinear response of SDOF systems [23].

The main parameters that govern the dynamic response of the 2SDOF system representing the main structure and NSC (Figure 1) can be easily derived from Equation (1). As matter of fact, the mass ratio $m_a/m$, as well as other parameters usually considered for simulating the response of SDOF systems can be identified as key parameters controlling the response of both the main structure and the NSC:

$$- \text{ Main structure}: \ T_1 = 2\pi\sqrt{\frac{m}{k}}; \ \xi = \frac{c}{2\sqrt{km}}; \tag{2}$$

$$- \text{ Non-structural component}: \ T_a = 2\pi\sqrt{\frac{m_a}{k_a}}; \ \xi_a = \frac{c_a}{2\sqrt{k_a m_a}}. \tag{3}$$

The elastic period T and the damping ratio $\zeta$ [22] defined in eqs. (2) and (3) for the main structure and the NSC, respectively, completely control the response of the 2DOF system in the linear-elastic range. Thus, a linear time-history analysis performed for a given seismic record the allows to evaluate both the maximum inertial force on the main structure $F_{el}$ and the one induced on NSC $F_{a,el}$. Then, the elastic threshold $F_y$ denoting the yielding of the main structure (Figure 1) can be easily defined as a further quantity of relevance in this parametric analysis as it corresponds, in principle, to a certain value of the force reduction factor $R$,

$$F_y = \frac{F_{el}}{R}. \tag{4}$$

However, the yielding force of the NSCs could be defined in a similar way, but it is omitted in this study as the response of the NSC is kept in the linear range. Finally, the parameters defined above have been changed within the range of variation defined below:

- mass ratio　　　　　　　$m_a/m \in \{0.01; 0.001\}$;
- main structure period　$T_1 \in [0.2 \text{ s}; 2.0 \text{ s}]$;
- secondary period　　　$T_a \in [0.1 \text{ s}; 5.0 \text{ s}]$;
- force-reduction factor　$R \in [1; 6]$.

Otherwise, both damping ratios $\zeta$ and $\zeta_a$ referred to the main structure and NSC have been assumed constant and equal to 0.05. As one can see, the considered mass ratios refer to a class of NSCs (such as systems, ceilings, etc.) whose mass is significantly lower (and fairly negligible) with respect to the structural one. The values of period $T_1$ are intended to cover the whole range of low-medium rise buildings, either made of steel or concrete, meanwhile, the values assumed for $T_a$ are intended to cover a wide spectrum of NSCs and their connections to the main structures, ranging from very stiff (and rigidly connected) components to fairly soft (or flexibly connected) ones. Finally, the values of $R$ range from non-dissipative structures ($R = 1$) up to highly dissipative ones ($R = 6$, simulating high ductility steel frames).

## 3. Results of the Parametric Analysis

The parametric analysis has been performed considering the 264 accelerograms [23] and the range of variation of the relevant parameters listed in Section 2. Consequently, 142560 nonlinear dynamic analyses have been run on 2DOF systems considering five values of $T_1$ (ranging between 0.2 s and 2.0 s) and nine for $T_a$ (between 0.1 s e 5.0 s). Two mass ratios (0.01 and 0.001) and six values of $R$ (from 1 to 6) have been also considered.

Figure 2 depicts the behavior of the ratio between the maximum absolute acceleration $F_a/m_a$ of NSC and the corresponding peak ground acceleration ($PGA = S \cdot a_g = \alpha \cdot S_g$) against the period ratio $T_a/T_1$.

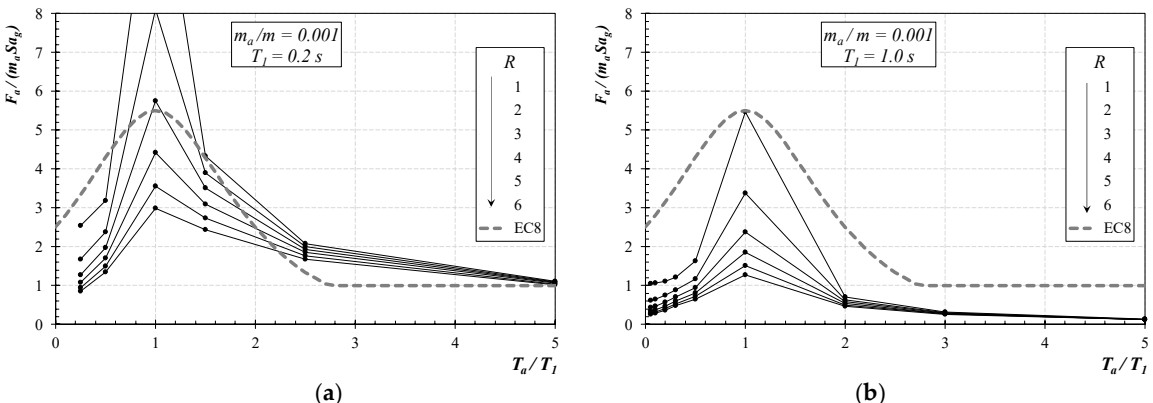

**Figure 2.** Peak absolute acceleration on the NSCs: $T_1 = 0.2$ s (**a**) and $T_1 = 1.0$ s (**b**).

The reduction factor $R$ ranges from 1 to 6 and each point is the average of the results obtained from the 264 seismic signals considered in the parametric study. Furthermore, the response of the code provision reported in EC8 [1] is also depicted resulting in a unique trend as such code formulation does not depends from the inelastic behavior of the main structure which is simulated by $R$ in this study. Specifically, Figure 2a refers to the case of main structures characterized by short period of vibration ($T = 0.2$ s) and demonstrates that the force reduction factor $R$ significantly affect the maximum ratios $F_a/m_a \cdot S \cdot a_g$ corresponding to the resonance condition ($T_a = T_1$), while the effect of $R$ results less important for long periods of NSC ($T_a/T_1 > 2$). Moreover, the simplified code provision reported in EC8 [1] miss this effect resulting in good agreement with numerical experiments only in the case of $R \approx 3 \div 4$, that is close to the values generally adopted in the seismic design of new RC structures.

A similar response is observed in Figure 2b in which the case of a medium-to-long-period of the main structure is represented. Furthermore, the maximum values of the ratio $F_a/m_a \cdot S \cdot a_g$, obtained for medium-to-long period structures (Figure 2b) are lower than the corresponding ones evaluated for short-period structures represented in Figure 2a. This effect is due to the reduction in the acceleration induced on the main structure for long periods. However, the R factor still affects significantly the dynamic response of the NSC for systems with $T_a/T_1 < 2$, while the prediction based on EC8 [1] results too conservative especially for high values of $R$.

Moreover, the results obtained for the mass ratio $m_a/m = 0.01$ overlap the ones obtained for $m_a/m = 0.001$, pointing out that, in this range of values, the mass ratio has a negligible influence on the resulting response. Therefore, the results for $m_a/m = 0.01$ are omitted hereinafter for sake of brevity.

As a final remark, as it can be easily understood through elementary mechanical intuition, Figure 2 shows that the two parameters $T_1$ and $R$ play a fundamental role in determining the peak absolute acceleration $S_a$ of NSCs. Therefore, this couple of parameters should be considered as the most influential ones in order to improve the relationships currently available for evaluating the dynamic response of non-structural components [1] which generally does not take into account the effect of the force reduction factor $R$.

### 3.1. Definition of Relevant Response Parameters

The results reported Figure 2 and, specifically, the comparison with the simplified formula adopted in EC8 [1] point out the significant lack of predictive capability affecting the aforementioned seismic code. As matter of fact, a wider number of parameters should be considered with the aim of enhancing the accuracy of formulations currently available. Moreover, more consistent response parameters can also be defined for describing the dynamic response of NSCs. For this purpose, [20,21] defined the following two parameters

$$\text{AmplificationFactor}: \quad AF = \frac{F_a(R; T_1, T_a/T_1, m_a/m; \xi, \xi_a)}{F_a(R = 1; T_1, T_a/T_1, m_a/m; \xi, \xi_a)}; \tag{5}$$

$$\text{ResonanceFactor}: \quad RF = \frac{F_a(R; T_1, T_a/T_1, m_a/m; \xi, \xi_a)/m_a}{F_r(R; T_1, T_a/T_1, m_a/m; \xi, \xi_a)/m}. \tag{6}$$

AF is the ratio of the peak total acceleration of the NSC evaluated for an inelastic main structure and the corresponding one derived by considering an elastic behavior of the latter, whereas RF is the ratio between the maximum total acceleration of the NSC and the maximum value of the total acceleration in the main structure.

In the following two subsections, the variation of the aforementioned parameters is deeply analyzed against the properties which fully describe the dynamic response of the system.

### 3.1.1. Amplification Factor

The amplification factor (AF) is analyzed and plotted against the period of the NSC for values of the factor $R$ ranging from 1 to 6 and a given period $T_1$. Specifically, Figure 3 reports this diagram for

the case of $m_a/m$=0.001 for two values of $T_1$ (namely, 0.2 and 1.0 s) and confirms the non-monotonic shape of the curves already described in the literature [21]. Moreover, it highlights once again the important role played by $R$ (especially when NSCs have low period of vibration) that is completely neglected by the current code formulations.

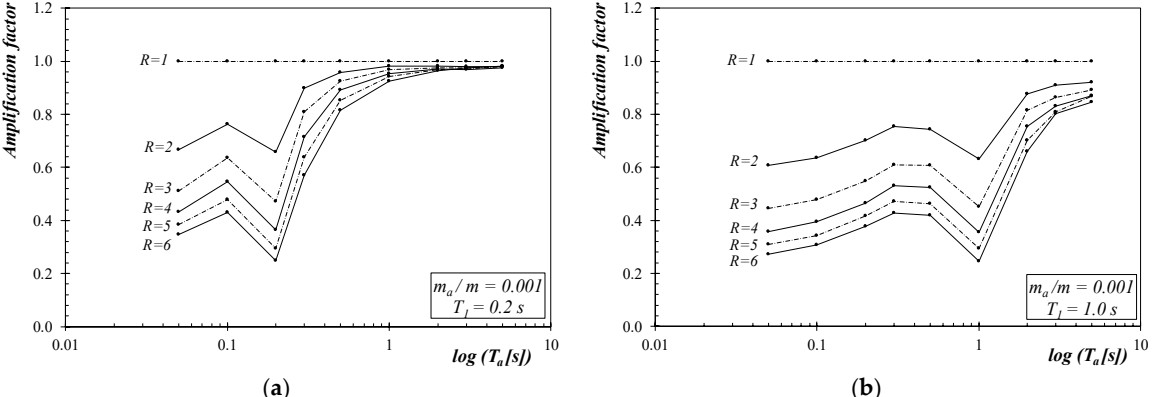

**Figure 3.** Amplification factor vs. the $T_a$: $T_1$ = 0.2 s (**a**) and $T_1$ = 1.0 s (**b**).

Moreover, Figure 4 consists of two diagrams reporting AF for the same fundamental period $T_1$ = 0.5 s and two different mass ratio $m_a/m$. It confirms that mass ratio is almost irrelevant for the resulting response, at least if it is kept lower than 0.01.

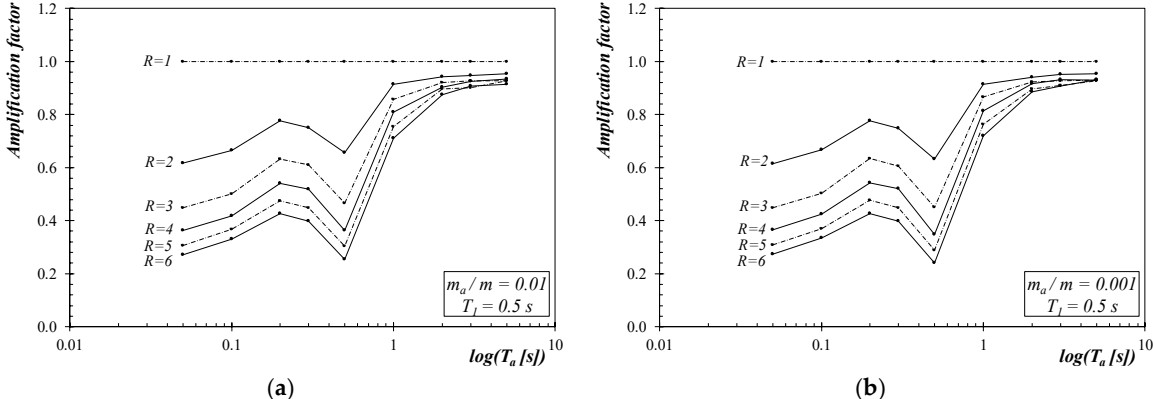

**Figure 4.** Amplification factor vs. the $T_a$: (**a**) $m_a/m$ = 0.01 and (**b**) $m_a/m$ = 0.001.

3.1.2. Resonance Factor

The resonance factor (RF) can obtain a more compact and representative representation of the huge amount of numerical results obtained in the parametric analysis herein performed. As shown in Equation (6), the denominator of RF is clearly related to the elastic spectral pseudo-acceleration of the main structure for the period $T_1$ and the damping ratio ξ, thus the possible analytical description of RF in terms of the other relevant parameters would straightforwardly lead to the quantification of $F_a$ which is the numerator in Equation (6).

The following figures report the trend obtained for RF by the NLTH analyses. It is worth noting that each point represents the average of 264 values derived by considering the set of seismic signals considered.

Figure 5 reports the (mean) values of RF for the cases of $T_1 \in$ {0.2 s; 0.5 s} and mass ratio equal to 0.001: $T_a/T_1$ ratio ranges up to significantly high values, owing to the short period of the main structure and the assumed range of variation of the secondary system periods (Section 2). Therefore, the curves (one for each value of the R factor) clearly highlight the following "properties" of RF:

- all the curves departs from 1 at $T_a/T_1 = 0$, which is a direct consequence of the resonance factor's definition (Equation 5);
- an almost linear branch connects 1 on the *y*-axis with the maximum value of RF (hereafter referred to as $RF_{max}$, in the following) that is due to a resonance between the two masses and is almost unaffected by *R* (at least for *R* > 2);
- RF tends to vanishes as $T_a/T_1 \to \infty$.

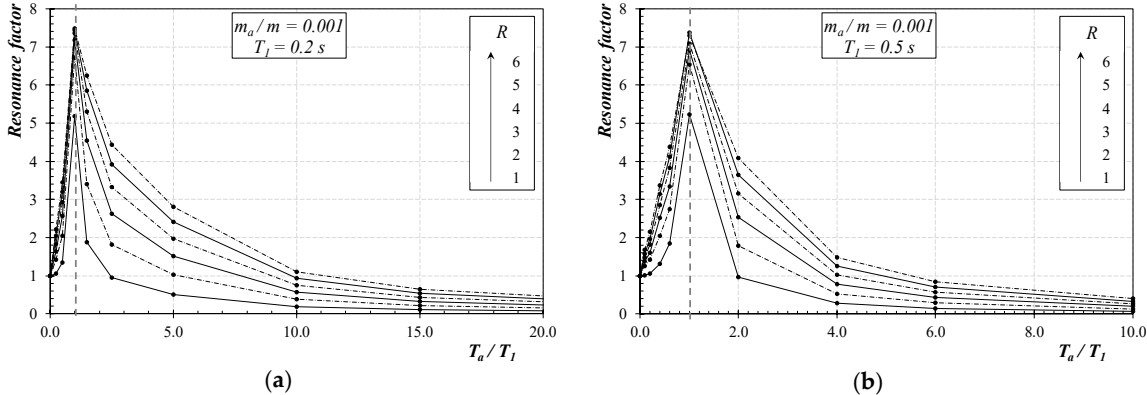

**Figure 5.** Mean value of RF vs. period ratio $T_a/T_1$ for $m_a/m = 0.001$: (**a**) $T_1 = 0.2$ s and (**b**) $T_1 = 0.5$ s.

Similar shapes are represented in Figure 6 for longer periods and in Figures 7 and 8 that report the same results for the higher mass ratio $m_a/m = 0.01$.

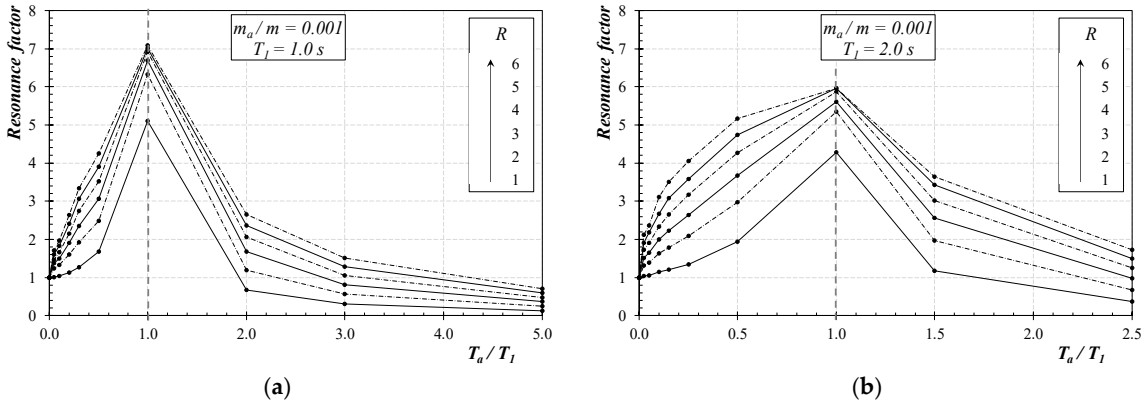

**Figure 6.** Mean RF vs. $T_a/T_1$ for $m_a/m = 0.001$: (**a**) $T_1 = 1.0$ s and (**b**) $T_1 = 2.0$ s.

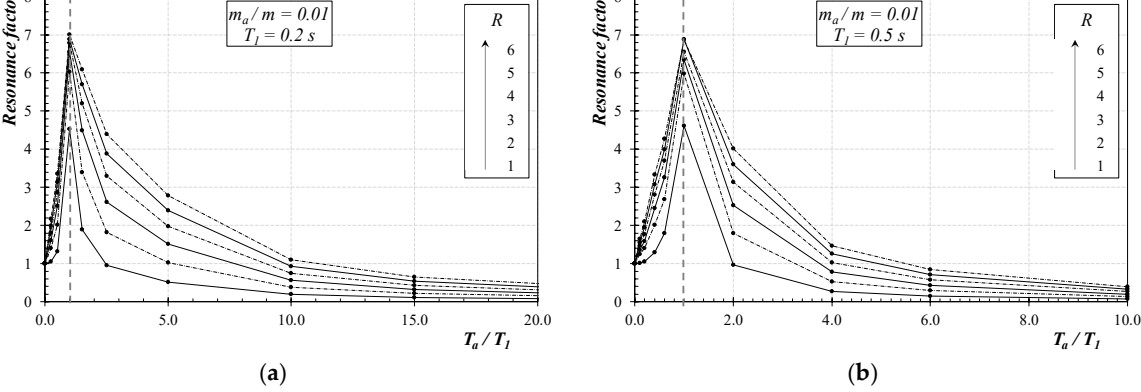

**Figure 7.** Mean RF vs. $T_a/T_1$ for $m_a/m = 0.01$: (**a**) $T_1 = 0.2$ s and (**b**) $T_1 = 0.5$ s.

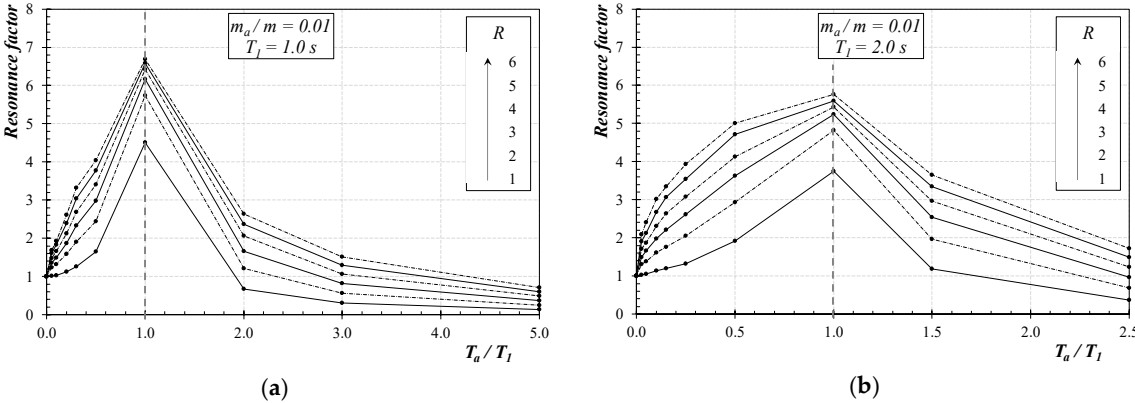

**Figure 8.** Mean RF vs. $T_a/T_1$ for $m_a/m = 0.01$: (**a**) $T_1 = 1.0$ s and (**b**) $T_1 = 2.0$ s.

Finally, since only the mean values of the main numerical results have been represented in the previous figures, further aspects dealing with the record-to-record variability of the RF need to be addressed. Therefore, Figure 9 reports both cumulative distribution and standard deviation of the RF obtained for the structural systems analyzed in this study. As for the former, the curves (conforming to a lognormal distribution) plotted in Figure 9 show that the R only influences the median value of RF. As for the latter, the standard deviation is mainly controlled by the period ratio $T_a/T_1$.

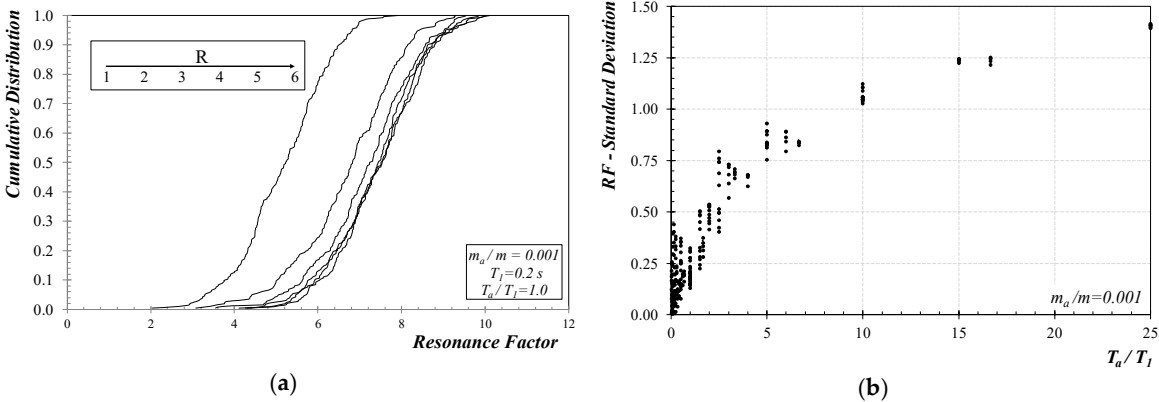

**Figure 9.** Record-to-record variability of the Resonance Factor: (**a**) Cumulative Distribution and (**b**) Standard deviation.

## 4. Concluding Remarks

This paper dealt with simulating the seismic response of "acceleration-sensitive" NSCs and determining the peak accelerations induced by the seismic shaking at the base of the main structure. The results of the wide parametric analysis reported in Section 3 can be summarized as follows:

- as already demonstrated in a recent paper [12], the available code provisions often neglect relevant parameters that, in fact, play a relevant role in controlling the response of "acceleration-sensitive" NSCs;
- specifically, the nonlinear behavior of the main structure, yet neglected in the current code provisions, plays a major role in modifying the actual seismic input on NSCs and, hence, determining its peak acceleration (and force) values;
- the definition of the resonance factor (RF) is a key step in quantifying the maximum seismic-induced actions;
- RF is clearly related with the other relevant parameters; specifically, the period $T_1$ of the main structure, the $T_a/T_1$ ratio and the reduction factor $R$ play a significant role in influencing the average value of *RF* determined from the nonlinear analyses;

- the standard deviation of the Resonance Factor RF is strictly correlated with by the period ratio $T_a/T_1$: higher dispersion affects the response of NSCs when they are (relatively) flexible.

Although further and more accurate calibrations might be proposed in the future, in the authors' opinion, relating the non-structural response to the structural one is the key conceptual contribution of this study.

**Author Contributions:** Conceptualization, E.M.; Methodology, E.M.; Software, E.M.; Validation, C.L.; Formal Analysis, E.M.; Investigation, C.L.; Resources, E.M.; Data Curation, C.L.; Writing—Original Draft Preparation, C.L. & E.M.; Writing—Review & Editing, C.L. & E.M.; Visualization, C.L.; Supervision, E.M.; Project Administration, E.M.; Funding Acquisition, E.M.

**Funding:** This study is part of the DPC-ReLUIS 2014–2018 Research Project whose financial support is gratefully acknowledged.

**Acknowledgments:** The Author wishes to acknowledge the contribution of Claudio Malangone who worked on the numerical analyses reported in this paper within the framework of his M.Sc. Thesis in Civil Engineering at the University of Salerno.

**Conflicts of Interest:** The authors declare no conflict of interest.

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
