# Peer review of "Seismic Response of Acceleration-Sensitive Non-Structural Components in Buildings"

_buildings, doi:10.3390/buildings9010007_

Round 1

Reviewer 1 Report

Comments to the Author

I have reviewed the paper “Seismic Response of Acceleration-Sensitive Non-Structural Components in Buildings”. This is an interesting work that needs few revisions to be published in Buildings. Here are my comments.

1. Within Introduction, add some references dealing with seismic reliability-base design approach to predict the seismic response, such as:

Castaldo, P.Palazzo, B.Alfano, G.Palumbo, M.F. (2018) Seismic reliability-based ductility demand for hardening and softening structures isolated by friction pendulum bearings, Structural Control and Health Monitoring, 25(11),e2256.

2. Explain the criteria to select the 264 natural ground motion records.

3. Explain if the cumulative distribution functions are lognormal.

4. Please, revise some typos in the text.

Author Response

REVIEWER #1

I have reviewed the paper “Seismic Response of Acceleration-Sensitive Non-Structural Components in Buildings”. This is an interesting work that needs few revisions to be published in Buildings. Here are my comments.

The authors wish to thank the Reviewer for both his/her positive evaluation and constructive suggestions.

1. Within Introduction, add some references dealing with seismic reliability-base design approach to predict the seismic response, such as:

Castaldo, P., Palazzo, B., Alfano, G., Palumbo, M.F. (2018) Seismic reliability-based ductility demand for hardening and softening structures isolated by friction pendulum bearings, Structural Control and Health Monitoring, 25(11),e2256.

A reference on this point has been actually added within the first paragraph of the introduction.

2. Explain the criteria to select the 264 natural ground motion records.

The 264 seismic signals are (basically) the same employed in a very well-known paper by Miranda (Miranda, 2010), ASCE Journal of Structural Engineering, 126(10), 1150-1158). They have been chosed with the aim to make analyses repeatable, as requested by any scientific "experiment".

3. Explain if the cumulative distribution functions are lognormal.

The cumulative curves represented in Figure 9 actually follow a lognormal distribution: this information has been added to the text.
However, neither in Figure 9, nor in other figures the analytical expressions of approximating functions are reported. This is a precise choice of the authors who, far from proposing the final expressions for relevant quantities, are interested in pointing out the main parameters influencing the response of acceleration sensitive non-structural components, which are nonetheless still neglected by the formulae adopted in current codes of standards.

4. Please, revise some typos in the text.

The manuscript has been carefully revised and some typing mistakes fixed.

Reviewer 2 Report

The paper addresses a very actual issue, which is related with the problem of the seismic response of non-structural elements. In the 2011 Lorca (Spain) Earthquake, most of the mortal victims where due to the fall of non-structural elements.

There are 21 references in the text, which are relatively new, but only about 19% from the last 5 years, and about 81% from the last 10 years.

Many parts of the present paper are almost equal to parts of the work presented in the paper corresponding to reference [12], namely it was adopted the same simplified 2 DOF system (Fig. 1 is similar), it was employed the same 264 natural records and Eq. (1) to (3) are also the same. Even the adopted mass ratio, main structure period, secondary period and force-reduction factor are equal, so it won’t be clear for a reader of both papers what are the real differences between them.

Moreover, I believe that the simplified 2DOF system is not suitable for capturing the seismic behaviour of masonry infill walls. However, as it is presented in lines 32 to 33, the reader might take the impression that the present study can be applied to this type of non-structural elements, which I believe it can’t, namely based on the results of many studies about the in-plane wall behaviour. See, for example, recent studies like:

https://www.sciencedirect.com/science/article/pii/S0141029617318527

I believe that the presented simplification can only capture the seismic response of elements like chimneys, parapets, gables, antennae, and so one.

Only after a major, major revision the paper would be suitable for publishing. The authors should emphasize the novelty of the present work, and what is the new contribution for the problem solution, namely by presenting new conclusions supported by new studies.

To support my opinion, see for example the first conclusion remark presented in this paper, which is:

 “the available code provisions are not sufficiently general and accurate in predicting the seismic response of NSCs: they generally neglect relevant parameters that, in fact, play a relevant role in controlling the response of such components;”

In reference [12] it is presented almost the same conclusion, which is:

“the available code provisions often neglect relevant parameters that, in fact, reveal an important role in controlling the seismic response of non-structural components;”.

Or: “RF is clearly related with the other relevant parameters; specifically, the period T1 of the main structure, the Ta/T1 ratio and the reduction factor R play a significant role in influencing the average value of RF determined from the nonlinear analyses;”

And in reference [12] is stated that: “both the elastic period T1 (and not only the ratio Ta/T1 that is considered in the main code provisions) and the resulting force reduction factor R influence the maximum accelerations attained in the seismic response”.

Author Response

REVIEWER #2

The paper addresses a very actual issue, which is related with the problem of the seismic response of non-structural elements. In the 2011 Lorca (Spain) Earthquake, most of the mortal victims where due to the fall of non-structural elements.

The authors wish to thank the Reviewer for both recognising the current relevance of the subject and proposing several constructive comments: in the authors’ opinion, the revised version is certainly clearer and “sharper” due to those comments.

There are 21 references in the text, which are relatively new, but only about 19% from the last 5 years, and about 81% from the last 10 years.

Three more papers, all published in the last 5 years, have been added to the reference list in the revised version of the manuscript (one of the previously available reference has been crossed out of the revised manuscript).

Many parts of the present paper are almost equal to parts of the work presented in the paper corresponding to reference [12], namely it was adopted the same simplified 2 DOF system (Fig. 1 is similar), it was employed the same 264 natural records and Eq. (1) to (3) are also the same. Even the adopted mass ratio, main structure period, secondary period and force-reduction factor are equal, so it won’t be clear for a reader of both papers what are the real differences between them.

Actually, ref [12] can be considered as a preliminary study: it mainly proposes an overview of the historical evolution of the formulae adopted in structural codes for predicting the maximum accelerations (or inertial forces) induced by the seismic shaking on “acceleration sensitive” NSCs. The second last paragraph of Section 1 (revised manuscript) explains i) the aim and scope of ref [12], ii) its minimal ovelap with the current paper and iii) the main novel aspects of this paper (which as highlighted in 3 bullet points).

Moreover, I believe that the simplified 2DOF system is not suitable for capturing the seismic behaviour of masonry infill walls. However, as it is presented in lines 32 to 33, the reader might take the impression that the present study can be applied to this type of non-structural elements, which I believe it can’t, namely based on the results of many studies about the in-plane wall behaviour. See, for example, recent studies like:

https://www.sciencedirect.com/science/article/pii/S0141029617318527

I believe that the presented simplification can only capture the seismic response of elements like chimneys, parapets, gables, antennae, and so one.

The authors agree that the 2DOF is not suitable for the “deformation sensitive” non structural components (e.g. masonry infills and partitions) considered in the recent paper by Sousa & Monteiro.

Conversely, several papers (among which refs. 20 and 21 in the revised manuscript) consider a 2DOF system with the aim to predict the seismic-induced accelerations on “acceleration-sensitive” NSCs, which are of relevance for the present work. A specific coment about i) the aforementioned classification of NSCs and ii) the fact that this study is limited to “acceleration-sensitive” ones (which is indeed even reported in the title) has been added in the last paragraph of Section 1.

Only after a major, major revision the paper would be suitable for publishing. The authors should emphasize the novelty of the present work, and what is the new contribution for the problem solution, namely by presenting new conclusions supported by new studies.

The revision of the introduction section makes clearer the actual novelties of the current paper and the main differences between ref [12] and this paper.

To support my opinion, see for example the first conclusion remark presented in this paper, which is:

 “the available code provisions are not sufficiently general and accurate in predicting the seismic response of NSCs: they generally neglect relevant parameters that, in fact, play a relevant role in controlling the response of such components;”

In reference [12] it is presented almost the same conclusion, which is:

“the available code provisions often neglect relevant parameters that, in fact, reveal an important role in controlling the seismic response of non-structural components;”.

Or: “RF is clearly related with the other relevant parameters; specifically, the period T1 of the main structure, the Ta/T1 ratio and the reduction factor R play a significant role in influencing the average value of RF determined from the nonlinear analyses;”

And in reference [12] is stated that: “both the elastic period T1 (and not only the ratio Ta/T1 that is considered in the main code provisions) and the resulting force reduction factor R influence the maximum accelerations attained in the seismic response”.

The role of some parameters (e.g., T1, R, ecc) currently neglected by the current code provisions was already figured out in ref [12], as the EC8 provisions were compared with the (average) results of NLTH analyses on the 2DOF system considered in both studies: therefore, the comments proposed in ref. [12] should be taken as “conjectures” about a natural phenomenon. Those “conjectures” are actually confirmed by the results of this study, where specific output parameters (e.g. “RF”, that is not considered in ref [12]) are introduced and considered. Moreover, the present study is not limited to the average values of output parameters, but it shows some features of their record-to-record variability. Far from saying the final word on the complex subject of predicting the response in “acceleration sensitive” NSCs, this paper shed a new light on the bunch of parameters affecting the aforementioned response and figures out some new information about its natural dispersion.

Round 2

Reviewer 1 Report

Comments to the Author

I have reviewed the revised paper “Seismic Response of Acceleration-Sensitive Non-Structural Components in Buildings”. The manuscript has been modified in compliance with the suggestions and so can be published in Buildings.

Reviewer 2 Report

I believe that the paper has been improved. It is now much clearer in the introduction what are the main objectives of the work and the boundaries for the use of the proposed simplified approach, namely by clarifying that the present study does not cover the case of “deformation-sensitive” non-structural elements, such as the in-plane behaviour of masonry infill walls.

In the conclusions, it is now more evident the contribution of the present work for the problem, which was not covered by the work presented in the reference [12].

The results are interesting and are in according with the new direction of seismic codes, such as the expression that I believe it will be presented in the next “Circolare” about the implementation of the new Italian seismic code NTC 2018 (with a proposed R factor being a function of Ta/Ti), so it is a good contribution for the necessary changes that must be introduced in the next generation of the Eurocode 8.

The paper is now acceptable for publishing.